# Progressive Multi-Modal Fusion for Robust 3D Object Detection

**Rohit Mohan**[1], **Daniele Cattaneo**[1], **Florian Drews**[2], **Abhinav Valada**[1]

[1] Department of Computer Science, University of Freiburg, Germany
[2] Bosch Research, Robert Bosch GmbH, Renningen, Germany
http://profusion3d.cs.uni-freiburg.de

**Abstract:** Multi-sensor fusion is crucial for accurate 3D object detection in autonomous driving, with cameras and LiDAR being the most commonly used sensors. However, existing methods perform sensor fusion in a single view by projecting features from both modalities either in Bird's Eye View (BEV) or Perspective View (PV), thus sacrificing complementary information such as height or geometric proportions. To address this limitation, we propose ProFusion3D, a progressive fusion framework that combines features in both BEV and PV at both intermediate and object query levels. Our architecture hierarchically fuses local and global features, enhancing the robustness of 3D object detection. Additionally, we introduce a self-supervised mask modeling pre-training strategy to improve multi-modal representation learning and data efficiency through three novel objectives. Extensive experiments on nuScenes and Argoverse2 datasets conclusively demonstrate the efficacy of ProFusion3D. Moreover, ProFusion3D is robust to sensor failure, demonstrating strong performance when only one modality is available.

**Keywords:** 3D Object Detection, Multimodal Learning, Self-Supervised Learning

## 1 Introduction

Reliable object detection is crucial for automated driving, as it enables vehicles to safely navigate the environment by accurately identifying and locating objects in their surroundings [1, 2]. To enhance robustness [3], LiDAR-camera fusion has emerged as the predominant paradigm, leveraging complementary information from different modalities. However, the different data distributions that stem from the inherently different nature of these modalities present a significant challenge. To address this limitation, various multi-modal fusion strategies [4, 5, 6, 7] have been explored, which mostly differ in the representation used for fusion: Bird's Eye View (BEV) or Perspective View (PV), and the stage at which fusion is performed: raw input, intermediate features, or object queries. While using a single representation eases the fusion by mapping the features of both modalities into a common space, it also leads to information loss such as height information in BEV and occlusions and perspective distortion in PV. Similarly, fusion at the raw input stage leads to the incorporation of sensor noise and irrelevant information. In intermediate feature fusion, modality-specific information can be lost, and object query-level fusion allows the integration of high-level semantic information but greatly relies on the quality of this information, potentially affecting robustness. In this paper, we propose a progressive fusion approach that fuses features from each modality in both BEV and PV at both intermediate features and object query levels. This incremental integration enables the model to effectively combine and leverage the strengths of different views and fusion stages.

Recently, self-supervised learning approaches such as Mask Image Modeling (MIM) [8, 9] and Mask Point Modeling (MPM) [10, 11] have shown great success in extracting strong representations from large-scale unlabeled data for images and point clouds, respectively. These methods enhance performance on fine-tuned downstream tasks and improve data efficiency [12]. This raises an important question: can similar techniques be applied to multi-modal sensor fusion in automotive

8th Conference on Robot Learning (CoRL 2024), Munich, Germany.

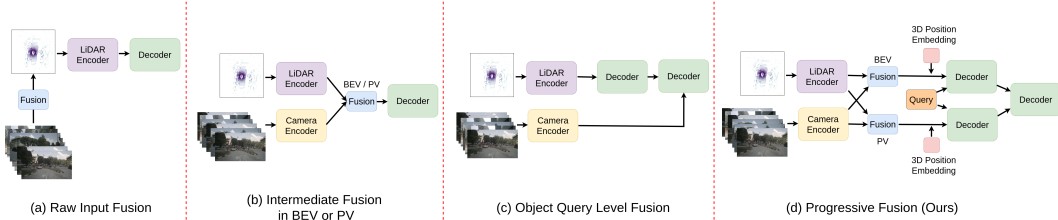

Figure 1: Overview of multi-modal fusion strategies. (a) Raw input fusion: integrates sensor data directly, prone to noise. (b) Intermediate fusion: combines features in BEV or PV, may lose modality details. (c) Object Query level fusion: merges high-level semantic information, reliant on its quality. (d) Progressive fusion (proposed): incrementally integrates features from BEV and PV at intermediate and object query levels.

settings for 3D object detection? While some prior works [13, 14] have explored pre-training with LiDAR-camera fusion on unlabeled datasets, they are often designed for small and dense point clouds with uniform point densities typical of indoor environments. However, these methods are not adaptable for automotive settings, which involve larger, sparser, and more heterogeneous data [15]. As a solution to this problem, we introduce an effective multi-modal mask modeling pre-training framework tailored for sparse outdoor point clouds.

In this paper, we propose the ProFusion3D framework, which performs progressive LiDAR-camera sensor fusion, and a novel self-supervised masked modeling pre-training strategy to enhance the performance of 3D object detection. Our model fuses features at two stages, the intermediate feature level and the object query level, across both BEV and PV. This comprehensive fusion strategy enables our model to effectively leverage both local and global features, significantly improving the accuracy and robustness of 3D object detection, even in the event of sensor failures. The main contributions of this work can be summarized as follows: (1) we propose ProFusion3D, a novel method for LiDAR-camera 3D object detection that performs fusion at different stages and in different views. (2) A novel inter-intra modality fusion module that performs joint self and cross window attention of view-specific features. (3) A self-supervised pre-training strategy for outdoor LiDAR-camera fusion methods. (4) Extensive evaluations and ablation study of ProFusion3D on nuScenes and Argoverse2 benchmarks.

## 2   Related Work

**Unimodal 3D Object Detection**: For camera-based 3D object detection, some methods [16, 17, 18] lift 2D features into 3D space via depth estimation or using frustum mesh grids, while others employ detection Transformers [19, 20, 21] to map image views to 3D space. LiDAR-based detectors convert point clouds into representations like voxels [22, 23], pillars [24], or range images [25], processing them akin to 2D detectors.

**Multi-modal 3D Object Detection:** LiDAR and camera fusion in 3D object detection have gained significant attention due to their superior performance. Existing 3D detection methods typically perform multi-modal fusion at one of three stages: raw input, intermediate feature, or object queries. Input fusion [4, 5] incorporates the 3D point cloud with 2D features from the camera. For example, PointPainting [4] uses category scores or semantic features [26, 27] from a 2D instance segmentation network. On the other hand, fusion at the intermediate feature level [6, 28, 29] maps both modalities into a shared representation space. Lastly, object query fusion methods [7, 30, 31, 32] either aggregate multi-modal features via proposals or refine queries from high-response LiDAR regions through interaction with camera features in two stages at the detection head. In this work, we seek to leverage the benefits of both intermediate feature fusion and object query fusion through a progressive fusion approach.

**Mask Image/Point Modeling:** Building on the success of masked language modeling methods [33, 34], similar techniques have been employed in the image domain [8, 9]. Among these, MAE stands out as a straightforward approach for masking random image patches and using their pixel values as reconstruction targets. Similarly, several methods have adapted these techniques to the point

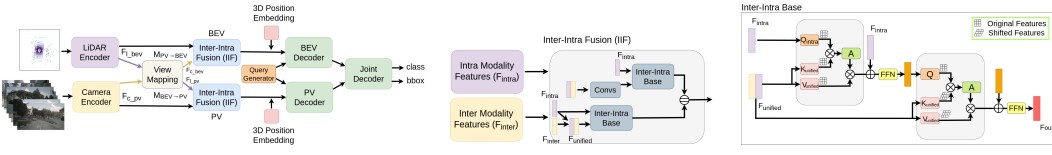

| (a) ProFusion3D Architecture | (b) Inter-Intra Fusion | (c) Inter-Intra Base Architecture |

Figure 2: (a) Illustration of our proposed ProFusion3D architecture that employs progressive fusion. (b) The topology of our proposed fusion module and (c) The core component of the aforementioned fusion module.

cloud domain. Point-BERT [10] and Point-MAE [11] utilize a BERT-style pre-training for point clouds by masking and reconstructing parts of the input. However, these methods generally focus on small dense point clouds. Voxel MAE [15] and Occupancy-MAE [35] adapt the MAE approach to handle sparse point clouds for 3D object detection which are common in outdoor autonomous driving scenarios. Despite extensive work on unimodal masked autoencoders, to the best of our knowledge, no approaches integrate LiDAR and multi-view camera modalities through multi-modal mask modeling in automotive settings. Our work bridges this gap by developing a framework to learn rich multi-modal representations for improved data efficiency. In particular, we introduce novel pre-training objectives of denoising and cross-modal consistency, increasing robustness and enforcing cross-modal information sharing.

## 3 Method

In this section, we first present the architecture of our proposed ProFusion3D framework that utilizes progressive fusion for 3D object detection, as shown in Fig. 2a, and then we detail its main components: the inter-intra fusion module and the decoder. Finally, we introduce our multi-modal masked modeling formulation for ProFusion3D (Fig. 3), and detail the three pre-training objectives.

### 3.1 ProFusion3D Architecture

Our ProFusion3D architecture takes LiDAR point cloud and multi-view camera images as input, encoding each with their corresponding encoders to compute $F_{\text{l\_bev}}$ BEV features for the LiDAR modality and $F_{\text{c\_pv}}$ PV features for camera modality. We refer to the modality native to a view as intra-modality. Thus, for the BEV space, $F_{\text{l\_bev}}$ is considered the intra-modality, whereas for the PV space, $F_{\text{c\_pv}}$ is the intra-modality. Next, we perform cross-view mapping by projecting $F_{\text{l\_bev}}$ to PV and $F_{\text{c\_pv}}$ to BEV, referring to these mapped features as the inter-modality features for each view space: $F_{\text{l\_pv}} = \mathcal{M}_{BEV \rightarrow PV}(F_{\text{l\_bev}})$ and $F_{\text{c\_bev}} = \mathcal{M}_{PV \rightarrow BEV}(F_{\text{c\_pv}})$. We then utilize our proposed inter-intra fusion block to fuse both modalities in each feature space. The resulting fused features $F_{\text{bev}} = IIF(F_{\text{l\_bev}}, F_{\text{c\_bev}})$ and $F_{\text{pv}} = IIF(F_{\text{c\_pv}}, F_{\text{l\_pv}})$ are fed to the decoder, which performs object query level fusion.

*LiDAR BEV to PV mapping $\mathcal{M}_{BEV \rightarrow PV}$:* For each pixel $(i_b, j_b)$ of the given LiDAR BEV features, we first compute the corresponding $(x, y)$ metric location. Next, we retrieve the $z$-coordinate for all non-empty voxels whose BEV projection falls into $(x, y)$. Finally, we project the resulting 3D points into the camera image coordinate frame $(i_c, j_c)$ using the camera extrinsic and intrinsic parameters.

*Camera PV to BEV mapping $\mathcal{M}_{PV \rightarrow BEV}$:* We "lift" [18] 2D camera PV features to BEV by predicting, for each camera pixel, a depth distribution over a fixed set of discrete depths. Using the camera intrinsic and extrinsic parameters, and the predicted depth distribution, we transform the image PV features into 3D to obtain the voxel grid $V \in \mathbb{R}^{X \times Y \times Z \times C}$. Following [6], we further encode these voxel features into BEV space features by first reshaping $V \in \mathbb{R}^{X \times Y \times (ZC)}$ and then applying 3D convolutions to reduce the channel dimension.

**Inter-Intra Fusion**: Our proposed inter-intra fusion block $IIF(F_{\text{intra}}, F_{\text{inter}})$ takes $F_{\text{intra}}$ and $F_{\text{inter}}$ features as its input, which are represented by $F_{\text{l\_bev}}$ and $F_{\text{c\_bev}}$ for the BEV space, and $F_{\text{c\_pv}}$ and $F_{\text{l\_pv}}$ for the PV space, respectively. Namely, for the BEV space, the LiDAR represents the intra-modality, and the camera represents the inter-modality, and conversely for the PV space. Fig. 2b illustrates the architecture of our fusion block, which utilizes two inter-intra base units.

The first inter-intra base unit applies two subsequent layers of window attention followed by a Feed-Forward Network (FFN) with residual connections, as shown in Fig. 2c. More precisely, it concatenates $F_{\text{inter}}$ and $F_{\text{intra}}$ to obtain $F_{\text{unified}}$, and then partitions the feature map into non-overlapping windows of size $M \times M$ for both $F_{\text{unified}}$ and $F_{\text{intra}}$. In this process, $F_{\text{unified}}$ is used to obtain the keys ($K_{\text{unified}}$) and values ($V_{\text{unified}}$) for the window attention, while $F_{\text{intra}}$ is used to get the queries ($Q_{\text{intra}}$). Following [36], we add a relative position bias $B \in \mathbb{R}^{M^2 \times M^2}$ to provide positional information. Applying attention with these key, query, and value matrices between the corresponding partitions effectively constitutes window attention, and can be defined as

$$\text{Attention}(Q_{\text{intra}}, K_{\text{unified}}, V_{\text{unified}}) = \text{Softmax}\left(\frac{Q_{\text{intra}} K_{\text{unified}}^T}{\sqrt{d}} + B\right) V_{\text{unified}}, \tag{1}$$

where $d$ is the embedding dimension. Afterward, the $F_{\text{unified}}$ features are shifted by $\left[-\left\lfloor\frac{M}{2}\right\rfloor, \left\lfloor\frac{M}{2}\right\rfloor\right]$ to obtain the key and value for another layer of window attention. The output of the previous window attention, passed through an FFN layer, acts as the query for this new window attention layer. By shifting the key and value features, while keeping the query features fixed, we can create a cross-window connection, effectively increasing the cross-modal attention span. We then apply the attention mechanism defined in Eq. (1), followed by another FFN layer.

The second inter-intra base unit is analogous to the first unit, but with the main difference being that the fused features $F_{\text{unified}}$ are first processed through a series of convolutional layers before being passed to the second unit. This approach allows the second unit to capture globally enriched spatial context, while the first unit focuses on capturing local context. We concatenate the outputs of these units to form the final output of the fusion module.

Overall, our fusion block effectively integrates features from both modalities, conditioned on intra-modal features, to yield comprehensive fused features. This hierarchical approach strikes a balance between capturing local patterns—crucial for identifying and distinguishing objects—and gaining a broader contextual understanding of the scene and the relationships between objects. Note that we employ an instance of our fusion block in both the BEV and PV to fuse the view-specific features.

**Decoder**: The decoder of our network employs two parallel sets of standard transformer decoders [37] to interact with the fused features in BEV and PV individually. Each set consists of two decoder layers. The updated queries from these individual view-specific decoders are then merged and passed through two additional decoder layers, where the queries interact with both fused BEV and PV features simultaneously.

*3D position-aware features:* Before passing the features extracted in PV and BEV space to the decoder, we enhance them by encoding 3D positional information. We first generate an evenly-spaced meshgrid $G_i$ for each camera $i$, as well as a BEV meshgrid $G_{bev}$. Similar to [20], we define $G_i \in \mathbb{R}^{W_f \times H_f \times D_f}$ in the frustum space of the respective camera, where $W_f$ is the width, $H_f$ is the height, and $D_f$ is the depth of the frustum. Each point $a$ in the meshgrid can then be expressed as $G_i^a = (u_a \cdot d_a, v_a \cdot d_a, d_a, 1)^T$, where $(u_a, v_a)$ represent pixel coordinates in the $i$-th camera's PV, and $d_a$ denotes the depth value. Finally, the corresponding 3D coordinates $p_i^{3d}$ can be computed using the extrinsic calibration between the respective camera and the LiDAR sensor. Similarly, any point $b$ in the BEV meshgrid is expressed as $G_{bev}^b = (i_b, j_b, h_b, 1)$, where $(i_b, j_b)$ is the respective pixel location in the BEV feature space, and $h_b$ is the height value. The corresponding 3D point can be simply computed by multiplying the BEV pixel location with the BEV grid size $(s_x, s_y)$ as $p_{bev}^{3d} = (i_b \cdot s_x, j_b \cdot s_y, h_b, 1)$. These 3D points, representing each view, are then fed into an MLP network and transformed into 3D Positional Embeddings (PE). The fused features $F_{\text{bev}}$ and $F_{\text{pv}}$ are processed by a $1 \times 1$ convolution layer and subsequently added to the 3D PE of the respective view to generate the 3D position-aware features $F_{\text{bev}}^{3D}$ and $F_{\text{pv}}^{3D}$.

*Query Initialization:* Similar to [20], we first initialize a set of learnable anchor points in normalized 3D world coordinates with a uniform distribution. The coordinates of these 3D anchor points are then fed to a small MLP network with two linear layers to generate the initial object queries $Q_0$.

*Query Refinement:* We refine the initial object queries $Q_0$ independently in each view to obtain the refined queries $Q_{\text{bev}}$ and $Q_{\text{pv}}$ using two DETR-style decoder layers. We then concatenate

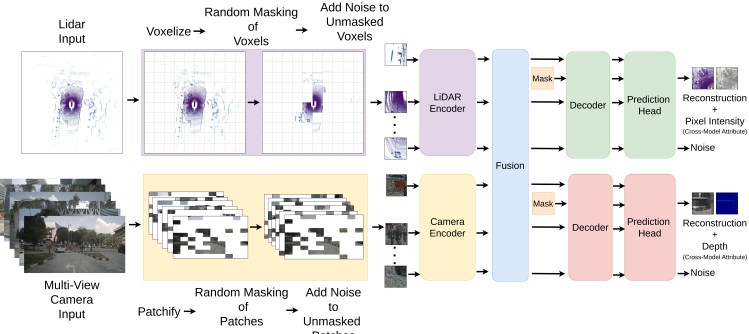

Figure 3: Illustration of our multi-modal mask modeling pipeline for learning multi-modal latent representations. It patchifies/voxelizes the input modalities into tokens, masks them asymmetrically, adds noise to the unmasked tokens, and trains the model with objectives of reconstruction, denoising, and cross-modal attribute prediction.

the view-specific branches to obtain the joint queries $Q_{\text{join}} = [Q_{\text{bev}}; Q_{\text{pv}}]$, and the joint features $F_{\text{join}}^{3D} = [F_{\text{bev}}^{3D}; F_{\text{pv}}^{3D}]$. We employ two additional decoder layers to predict the final object classes $\hat{p}$ and bounding boxes $\hat{b}$. We describe further details in the supplementary material. For training, we utilize the Hungarian algorithm [38] to find the optimal bipartite matching between the predicted objects and the ground truth objects as described in [37]. We adopt the focal loss for classification and L1 loss for 3D bounding box regression, following the approach in [20].

## 3.2 LiDAR and Multi-View Camera Mask Modeling

For the multi-modal masked modeling pre-training, we first patchify the images from multi-view cameras and voxelize the point clouds from LiDAR. For simplicity, we refer to them as tokens, using a modality-agnostic term. We then apply an asymmetrical masking strategy to randomly mask the tokens in each modality, where one modality is randomly selected to have a higher masking rate than the other. Next, we add Gaussian noise to the unmasked tokens. We then add positional encoding to these tokens, and the encoder of each modality processes them. Following this, our inter-intra fusion module performs multi-modal fusion, and simple decoders are then added on top to learn three pre-training objectives. Fig. 3 illustrates our pipeline for multi-modal masked modeling.

*Masked Token Reconstruction:* We reconstruct each masked image patch from the PV branch and voxels from the BEV branch to emphasize the learning of spatial correlations. For the former, we employ L1 loss between the predicted values of the masked pixels and the corresponding RGB values, similar to [9]. For the latter, we supervise it with Chamfer distance [11], which measures the distance between two sets of points.

*Unmasked Token Denoising:* We incorporate a denoising objective for each unmasked image patch and voxel to enhance the learning of high-frequency components, such as fine details and surface variations. For the unmasked image patches, we employ L1 loss between the predicted noise and the actual noise added. Similarly, for the unmasked voxels, we compute the denoising loss analogously to Chamfer distance, where the ground truth represents the actual noise added to each point.

*Masked Token Cross-Modal Attribute Prediction:* We introduce a cross-modal attribute prediction for masked tokens to leverage complementary information from corresponding unmasked modality tokens. Using the aforementioned asymmetrical masking strategy, masked tokens in the modality with the high masking rate will have a higher probability of the corresponding points in the other modality being unmasked. This setup allows us to create an objective that explicitly encourages the model to leverage complementary cross-modal information from the unmasked tokens to predict the attributes of the masked token. Specifically, we add a prediction task for the corresponding pixel intensity of the points in the masked voxels. For masked image patches, we predict the depth values. This ensures more effective multi-modal representation learning by leveraging cross-modal synergies. For training depth, we use the depth loss defined in [39]. For pixel intensity prediction, we add an L1 loss term to the Chamfer distance loss of masked token reconstruction. We outline each of the loss

terms for the pre-training objectives in detail in the supplementary material. During pre-training, we sum all the individual losses to compute the total pre-training loss.

## 4 Experimental Evaluation

**Dataset and Evaluation Metrics**: We perform comprehensive experiments on the large-scale nuScenes dataset [40] for 3D detection. This dataset provides point clouds from a 32-beam LiDAR and images with a resolution of $1600 \times 900$ from 6 surround cameras. We use mean average precision (mAP) and the nuScenes detection score (NDS) [40] as evaluation metrics. Additionally, we report mAP for the Argoverse2 dataset, which offers a long perception range of up to $200\,\mathrm{m}$, covering an area of $400\,\mathrm{m} \times 400\,\mathrm{m}$, significantly larger than nuScenes.

**Implementation Details**: We implement our network in PyTorch using the open-source MMDetection3D [41], with Swin-T [36] as the image backbone and VoxelNet [23] as the LiDAR backbone. FPN [42] fuses multi-scale camera features into a 1/8 resolution feature map. Camera images are downsampled to $448 \times 800$ and the LiDAR point cloud is voxelized at a resolution of $0.075\,\mathrm{m}$. Our fusion module uses a window size of $M = 7$. Our training comprises two stages: pre-training and 3D object detection training. For pre-training, we utilize the AdamW optimizer [43] configured with $\beta_1 = 0.95$, $\beta_2 = 0.99$, and a weight decay of 0.01. The learning rate is initially set to $5 \times 10^{-5}$, gradually increases to $5 \times 10^{-4}$ over the first 1000 iterations, and then decreases to $1 \times 10^{-7}$ according to a cosine annealing schedule. We use a batch size of 4 and run for 200 epochs, employing an asymmetric masking ratio of 0.7 and 0.3. For 3D object detection, we train our network for 24 epochs with CBGS [44] and a batch size of 16. The AdamW optimizer is employed with an initial learning rate of $1.0 \times 10^{-4}$ following a cyclic learning rate policy [45]. For details on loss, please refer to the supplementary material. Lastly, on the Argoverse2 dataset, we directly train for 3D object detection for 6 epochs similar to [46, 47].

### 4.1 Benchmarking Results

In Tab. 1, we compare our proposed approach with other state-of-the-art methods on the nuScenes (Tab. 1a) and Argoverse 2 (Tab. 1b) datasets for 3D object detection. Additionally, for the nuScenes dataset, we evaluate the robustness of ProFusion3D (Tab. 1c), investigating how the performance is affected when either the LiDAR or camera inputs are unavailable. Our method achieves 71.1% mAP on nuScenes and 37.7% mAP on Argoverse2, outperforming the best-performing baseline methods UniTR and CMT by 0.6% and 1.6%, respectively. This consistent improvement in performance across both datasets demonstrates that using progressive fusion minimizes the loss of information compared to single-stage or single-view methods, while multi-modal mask modeling enables learning of richer multi-modal representation. For the robustness evaluation presented in Tab. 1c, we train our model with the same robustness training strategy as described in [32]. Among fusion-based methods, in the event of a camera sensor failure, our method outperforms BEVFusion by 0.9%. In the case of LiDAR failure, it surpasses CMT by 0.7%. While baseline fusion methods show variable robustness with different sensor failures, our approach consistently maintains higher performance in each scenario and remains relatively closer to state-of-the-art unimodal methods.

### 4.2 Ablation Study

We first investigate the impact of our multi-modal masked modeling pipeline on the downstream 3D object detection task with regard to data efficiency and evaluate each pre-training objective. Subsequently, we discuss the impact of various components of our progressive fusion approach.

**Data Efficiency:** One of the major benefits of using self-supervised learning is a reduced need for annotated data. To study the effects of various dataset sizes, we train two versions of our model with varying subsets of the annotated dataset held out, where one model is initialized randomly and the other has been pre-trained on the entire dataset using our self-supervised objectives. Please refer to the supplementary material for the strategy of compilation of the subsets of the dataset. The results of this experiment are shown in Tab. 2a. The model trained with 80% of the data using our

Table 1: Performance comparison with state-of-the-art methods on nuScenes and Argoverse2 datasets. 'C' denotes Camera, and 'L' denotes LiDAR

(a) Evaluation on the nuScenes validation set.

| Methods | modality | mAP | NDS |
|---|---|---|---|
| DETR3D [19] | C | 34.9 | 43.4 |
| BEVDet4D [48] | C | 42.1 | 54.5 |
| BEVFormer [21] | C | 41.6 | 51.7 |
| SECOND [23] | L | 52.6 | 63.0 |
| CenterPoint [49] | L | 59.6 | 66.8 |
| FSD [50] | L | 62.5 | 68.7 |
| VoxelNeXt [46] | L | 63.5 | 68.7 |
| LargeKernel3D [51] | L | 63.3 | 69.1 |
| Transfusion-L [7] | L | 64.9 | 69.9 |
| FUTR3D [31] | LC | 64.5 | 68.3 |
| UVTR [28] | LC | 65.4 | 70.2 |
| TransFusion [7] | LC | 67.5 | 71.3 |
| BEVFusion [6] | LC | 68.5 | 71.4 |
| DeepInteraction [30] | LC | 69.9 | 72.6 |
| DAL [52] | LC | 70.0 | 73.4 |
| CMT [32] | LC | 70.3 | 72.9 |
| FSF [47] | LC | 70.4 | 72.7 |
| UniTR [53] | LC | 70.5 | 73.3 |
| **ProFusion3D (ours)** | LC | **71.1** | **73.6** |

(b) Benchmarking on the Argoverse2 validation set.

| Methods | Modality | mAP | CDS |
|---|---|---|---|
| CenterPoint [49] | L | 22.0 | 17.6 |
| FSD [50] | L | 28.2 | 22.7 |
| VoxelNeXt [46] | L | 30.5 | 23.0 |
| FSF [47] | LC | 33.2 | 25.5 |
| CMT [32] | LC | 36.1 | 27.8 |
| **ProFusion3D (ours)** | LC | **37.7** | **29.1** |

(c) Robust performance comparison on the nuScenes val set with missing modality in mAP.

| Methods | Modality at Inference | | |
|---|---|---|---|
| | Both | only LiDAR | only Cams |
| BEVFormer [21] | - | - | 41.6 |
| TransFusion-L [7] | - | 64.9 | - |
| BEVFusion [6] | 68.5 | 63.0 | 32.0 |
| CMT [32] | 70.3 | 61.7 | 38.3 |
| **ProFusion3D (ours)** | **71.1** | **63.9** | **38.9** |

multi-modal masking initialization outperforms the model trained with 100% of the data. The most significant performance increase is seen with the smallest subset of labeled data, demonstrating that low-data settings benefit the most from the learned multi-modal latent representations. Moreover, all pre-trained models consistently outperform their randomly initialized counterparts across all percentages of labels.

**Role of Pre-Training Objectives:** To investigate the synergy of the pre-training objectives, we pre-trained our model with different combinations of these objectives, focusing on the subset with 20% annotated data. The results, presented in Tab. 2b show that training with the masked token reconstruction objective alone provides a solid baseline, demonstrating effective multi-modal representations through the fusion module. Adding the unmasked token denoising objective yields a 0.8% gain in mAP due to improved robustness to noise, resulting in better fine detail extraction. The cross-modal token attribute prediction results in a 1.1% gain in mAP, further enhancing the model's ability to leverage features of complementary modalities. Combining both objectives achieves a 1.5% gain in mAP, demonstrating that the synergy of these pre-training objectives improves model performance by enhancing both robustness and cross-modal understanding.

**Impact of Fusion in Both View Spaces:** Tab. 2c presents the results of fusion in only the BEV space versus both BEV and PV spaces. Our inter-intra fusion (IIF) module in the BEV space achieves a 4.8% gain in mAP over the non-fusion counterpart, demonstrating its effectiveness. Performing fusion in both BEV and PV spaces results in an additional 1% gain in mAP, indicating that using both view spaces helps preserve more information compared to using a single view space.

**Importance of Inter-Intra Fusion:** To study the impact of our inter-intra attention, we replaced the fusion module with various configurations, as presented in Tab. 2d. Initially, we concatenated inter- and intra-features in each view space as fusion. Next, we applied self-attention on intra-features and then used cross-attention with inter-features to merge them. Following, we used one inter-intra base unit (refer to Sec. 3) as the fusion block. We then experimented using two successive inter-intra base units without concatenating inter- and intra-features for computing attention, where the first unit performed window self-attention on intra-features, and the second performed window cross-attention on inter- and intra-features. Finally, we used our inter-intra fusion (IIF) module. Our results show that the inter-intra base as the fusion block outperforms other variations while the IIF module achieves the best performance, underscoring the importance of both local and globally enriched spatial context.

**Need for View-Specific Decoders:** Tab. 2e compares the performance of a view-agnostic decoder that simultaneously interacts with fused view space features against our proposed configuration of

Table 2: Ablation results on the nuScenes dataset.

(a) 3D object detection performance for randomly and pre-trained initialization with varying labeled data. (Voxel Size = 0.15m, Image Size = 256 × 704)

| Labeled Data | Initialization | mAP |
|---|---|---|
| 20% | Random | 52.3 |
| | **Multi-Modal Masking** | **56.8** |
| 40% | Random | 59.5 |
| | **Multi-Modal Masking** | **63.2** |
| 60% | Random | 62.4 |
| | **Multi-Modal Masking** | **64.1** |
| 80% | Random | 64.1 |
| | **Multi-Modal Masking** | **65.6** |
| 100% | Random | 65.2 |
| | **Multi-Modal Masking** | **66.2** |

(b) Impact of pre-training objectives on initialization for 3D detection with 20% labeled data.

| Mask Token Reconstruction | Unmasked Token Denoising | Cross-Modal Attribute Prediction | mAP |
|---|---|---|---|
| ✓ | | | 55.3 |
| ✓ | ✓ | | 56.1 |
| ✓ | | ✓ | 56.4 |
| ✓ | ✓ | ✓ | **56.8** |

(c) Fusion in single BEV vs. both BEV and PV view space.

| Fusion | mAP |
|---|---|
| BEV LiDAR only | 64.7 |
| IIF in BEV | 69.5 |
| IIF in BEV + PV | **70.5** |

(d) Effect of different fusion variations.

| Fusion | mAP |
|---|---|
| Concatenation | 68.3 |
| Self- → Cross- Attention | 68.7 |
| Inter-Intra Base (ours) | 69.6 |
| Inter-Intra Base (Self- → Cross- Attention ) | 69.1 |
| IIF (ours) | **70.5** |

(e) View-agnostic decoder vs View-specific decoder.

| Fusion | mAP |
|---|---|
| View-agnostic decoder | 69.9 |
| View-specific decoder (ours) | **70.5** |

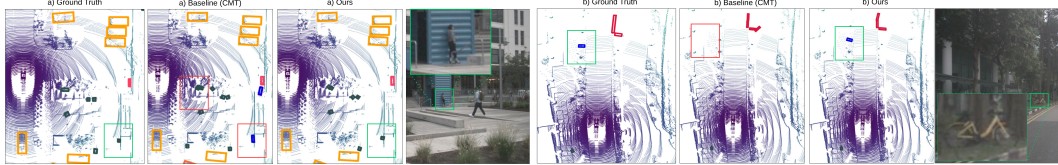

Figure 4: Qualitative 3D object detection results of our proposed ProFusion3D network versus the state-of-the-art CMT architecture on the nuScenes val set. Red: incorrect detections, Green: correct detection.

view-specific decoders followed by a view-agnostic decoder. The higher mAP in our configuration indicates that direct fusion via object queries causes an imbalance, with BEV dominating over PV.

## 5   Conclusion

In this work, we presented our ProFusion3D architecture for 3D object detection, which not only outperforms existing baselines on the nuScenes and Argoverse2 datasets but also demonstrates robustness in sensor failure scenarios. Our approach incorporates a progressive fusion strategy that integrates features from each modality in both BEV and PV space at the intermediate feature level and object query level. Furthermore, we introduced a self-supervised multi-modal mask modeling pre-training framework that significantly enhances ProFusion3D performance on small annotated datasets, thereby improving data efficiency. This framework employs masked token reconstruction with two novel objectives to facilitate robust multi-modal representation learning. Extensive evaluations demonstrate the efficacy of our ProFusion3D.

**Limitations:** Our method involves mapping from one view to another and thereby, relies on good calibration of sensors. To alleviate this dependency, we will explore learnable methods for alignment between views [54, 55]. Moreover, our model currently does not consider efficiency. In the future, we aim to develop a more advanced framework that uses uncertainty estimation [56] to dynamically select the most effective combination of different views and fusion levels, optimizing the balance between performance, efficiency, and robustness.

**Acknowledgments**

This research was funded by Bosch Research as part of a collaboration between Bosch Research and the University of Freiburg on AI-based automated driving.

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

# Progressive Multi-Modal Fusion for Robust 3D Object Detection
## *Supplementary Material*

In this supplementary material, we provide additional details on various aspects of our work. First, we provide further architecture details for our fusion module and decoder in Sec. 1. Next, we present the loss details of the three pre-training objectives for our proposed LiDAR and Multi-View Camera Mask Modeling in Sec. 2. In Sec. 3 we describe the strategy for selecting the subsets of the dataset used in our data efficiency experiment (Sec. 4.2 of the main manuscript). Lastly, we provide additional qualitative results for ProFusion3D in Sec. 4 and Sec. 5.

## 1    Additional Architectural Details of ProFusion3D

**Inter-Intra Fusion:** For each of the inter-intra base units, the corresponding channel dimension of the query, key, and value embeddings are set to 192. The following Feed-Forward Network (FFN) consists of two fully-connected layers with GeLU activation after the first one. The first layer expands the input channels to 1024 while the second layer condenses it back to the original embedding dimensions (192).

The convolutional block of the fusion module uses two consecutive $3 \times 3$ depthwise separable convolutions, each with dilation rates of 2 and 4, respectively. The number of channels in these convolutions is twice the embedding dimensions (384).

**Decoder:** We employ two DETR-style decoder layers in the BEV decoder, PV decoder, and Joint Decoder. We follow similar parameter settings as described in [1] for each of the decoder layers. The sequence of operations within each layer is as follows: self-attention, normalization, cross-attention, normalization, FFN, and normalization. The embedding dimensions are set to 256, and the FFN channels are set to 2048. Both the attention and FFN dropout rates are configured to 0.1. We set the number of queries to 600 for initializing $Q_0$. The initial object queries $Q_0$ interact with the 3D position-aware BEV features $F_{\text{bev}}^{3D}$ in the BEV decoder to update their representations to $Q_{\text{bev}}$. In parallel, $Q_0$ also interacts with the 3D position-aware PV features $F_{\text{pv}}^{3D}$ in the PV decoder to update their representations to $Q_{\text{pv}}$. Following this, $Q_{\text{join}} = [Q_{\text{bev}}; Q_{\text{pv}}]$ interacts with the 3D position-aware joint features $F_{\text{join}}^{3D} = [F_{\text{bev}}^{3D}; F_{\text{pv}}^{3D}]$ in the joint decoder to generate the final updated query representations.

We use two FFNs to predict the 3D bounding boxes and the classes using the updated queries in each of the decoder layers. The prediction for each decoder layer is then as follows:

$$\hat{b}_i^{\text{d}} = \phi^{\text{reg}}(Q_i^{\text{d}}), \quad \hat{p}_i^{\text{d}} = \phi^{\text{cls}}(Q_i^{\text{d}}) \tag{1}$$

where $\phi^{\text{reg}}$ and $\phi^{\text{cls}}$ represent the FFNs for regression and classification, respectively. $Q_i^{\text{d}}$ are the updated object queries of the $i$-th decoder layer of the $d$-th decoder, where $d \in \{\text{bev}, \text{pv}, \text{joint}\}$.

We train ProFusion3D through set prediction by using bipartite matching for one-to-one assignment between predictions and ground truths. Specifically, we use the focal loss for classification and L1 loss for 3D bounding box regression:

$$L(y, \hat{y}) = \lambda_1 L_{cls}(p, \hat{p}) + \lambda_2 L_{reg}(b, \hat{b}) \tag{2}$$

where $\lambda_1$ and $\lambda_2$ are the hyperparameters to balance the two loss terms.

8th Conference on Robot Learning (CoRL 2024), Munich, Germany.

## 2   Loss Functions for Pre-Training Objectives

For the multi-modal masked modeling, we introduce three pre-training objectives: masked token reconstruction, unmasked token denoising, and masked token cross-modal attribute prediction. To train each objective we employ the following losses:

**Masked Token Reconstruction:** In this pre-training objective, we reconstruct each masked image patch from the PV branch and voxels from the BEV branch. For the image patch reconstruction, we employ L1 loss between the predicted values of the masked pixels and the corresponding RGB values as follows:

$$L_{\text{L1}} = \frac{1}{N_{mp}} \sum_{i=1}^{N_{mp}} \left| \hat{I}_i - I_i \right| \tag{3}$$

where $\hat{I}_i$ and $I_i$ are the predicted and ground truth RGB value of the $i$-th masked pixel, respectively, and $N_{mp}$ is the total number of masked pixels.

For the voxel reconstruction, let $P_{\text{gt},i} = \{x_1, x_2, \cdots, x_N\}$ be the $i$-th masked voxel where $N$ is the number of fixed points in voxels and $P_{\text{rec},i} = \{\tilde{x}_1, \tilde{x}_2, \cdots, \tilde{x}_N\}$ be the corresponding reconstruction. Then the Chamfer loss is defined as follows:

$$L_{\text{Chamfer}}(P_{\text{gt,i}}, P_{\text{rec,i}}) = \frac{1}{|P_{\text{gt,i}}|} \sum_{x \in P_{\text{gt,i}}} f(x, P_{\text{rec,i}}) + \frac{1}{|P_{\text{rec,i}}|} \sum_{\tilde{x} \in P_{\text{rec,i}}} f(\tilde{x}, P_{\text{gt,i}}),$$
$$f(x, P) = \|x - P^j\|_2^2, \text{with } j = \arg\min_k \|x - P^k\|_2^2 \tag{4}$$

where $\| \cdot \|_2$ denotes the L2-norm. This loss function ensures that each point in the ground truth set $P_{\text{gt}}$ is close to some point in the reconstructed set $P_{\text{rec}}$ and vice versa.

**Unmasked Token Denoising:** In this pre-training objective, we learn to predict noise for each unmasked image patch and voxel. For the unmasked image patches, we employ L1 loss between the predicted noise and the actual noise added as follows:

$$L_{\text{denoise\_image}} = \frac{1}{N_{up}} \sum_{i=1}^{N_{up}} |\hat{n}_i - n_i| \tag{5}$$

where $\hat{n}_i$ is the predicted noise and $n_i$ is the actual noise added to the $i$-th unmasked pixel, and $N_{up}$ is the total number of unmasked pixels.

For the voxel denoising, let $N_{\text{a},i} = \{n_1, n_2, \cdots, n_N\}$ be the noise added to the $i$-th unmasked voxel and $N_{\text{p},i} = \{\tilde{n}_1, \tilde{n}_2, \cdots, \tilde{n}_N\}$ be the corresponding noise prediction.

$$L_{\text{denoise\_voxel}}(N_{\text{a,i}}, N_{\text{p,i}}) = \frac{1}{|N_{\text{a,i}}|} \sum_{n \in N_{\text{p,i}}} \min_{\tilde{n} \in N_{\text{p,i}}} \|n - \tilde{n}\|_2^2 + \frac{1}{|N_{\text{p,i}}|} \sum_{\tilde{n} \in N_{\text{n,i}}} \min_{x \in N_{\text{a,i}}} \|\tilde{n} - n\|_2^2. \tag{6}$$

**Masked Token Cross-Modal Attribute Prediction:** In this pre-training objective, we predict pixel intensities for points in masked voxels and depth values for masked image patches. To predict pixel intensities for points in masked voxels, we do this jointly with the masked voxel reconstruction. Hence, while predicting the points in the masked voxel, we also predict the corresponding pixel intensities. For pixel intensity prediction, we add a loss term to the Chamfer distance loss of masked token reconstruction, by replacing $f$ in Eq. (4) with $\tilde{f}$:

$$\tilde{f}(x, P) = \|x - P^j\|_2^2 + \lambda |x_I - P_I^j|, \text{with } j = \arg\min_k \|x - P^k\|_2^2 \tag{7}$$

where $\lambda$ is the loss balancing term, $x_I$ and $P_I^j$ are the pixel intensities of $x$ and of the $j$-th point in $P$.

Following, for depth prediction for masked image patches, let $d$ and $\hat{d}$ denote the ground-truth and the predicted depth, respectively. Our loss function for depth estimation is then defined by

$$L_{\text{depth}}(d, \hat{d}) = \frac{1}{N_{mp}} \sum_i^{N_{mp}} \left( \log d_i - \log \hat{d}_i \right)^2 - \frac{1}{N_{mp}^2} \left( \sum_i^{N_{mp}} \left( \log d_i - \log \hat{d}_i \right) \right)^2$$
$$+ \left( \frac{1}{N_{mp}} \sum_i^{N_{mp}} \frac{d_i - \hat{d}_i}{d_i} \right)^2. \tag{8}$$

This loss term is a combination of the scale-invariant logarithmic error and the relative squared error.

## 3  Strategy for Subset Selection in Data Efficiency Experiments

In Sec. 4.2 of our main manuscript, we perform a data efficiency experiment that utilizes subsets of 20%, 40%, 60%, and 80% from the 100% training annotated data of nuScenes. To select scenes for each subset of the training dataset, we sorted the dataset based on scene timestamps and then used a systematic sampling method. Specifically, we divided the scenes into five groups based on their indices and selected scenes according to these groups:

- For a 20% subset, we included all scenes from one group (i.e., those with indices where $i$ mod $5 = 0$).

- For a 40% subset, we included scenes from two groups (i.e., those with indices where $i$ mod $5 \in \{0, 2\}$).

- For a 60% subset, we included scenes from three groups (i.e., those with indices where $i$ mod $5 \in \{0, 2, 4\}$).

- For an 80% subset, we included scenes from four groups (i.e., those with indices where $i$ mod $5 \in \{0, 1, 2, 4\}$).

This systematic sampling method helps to minimize temporal dependencies between frames and ensures that the reduced datasets retain a similar level of diversity as the complete dataset.

## 4  Visualization

In Fig. 1, we provide additional qualitative results of our proposed ProFusion3D on the nuScenes dataset. The dataset encompasses urban road scenes with objects ranging from cars, trucks, bicycles, motorbikes, people, barriers, and cones in a very cluttered environment with occlusions. Despite these challenging conditions, our ProFusion3D consistently and accurately detects all these objects.

## 5  Generalization in Real-World Scenarios

In this experiment, we perform a real-world zero-shot generalization experiment to evaluate the performance of our proposed ProFusion3D model using our in-house autonomous vehicle as depicted in Fig. 2. We evaluate the multi-modal ProFusion3D model under the assumption that a camera sensor fails during operation, meaning that the model operated without camera images. For this experiment, we utilized the model trained on the nuScenes dataset. It is important to note that our setup includes an Ouster 128 LiDAR, while nuScenes uses a Velodyne HDL32E. Thus, this experiment introduces additional challenges for object detection due to variations in sensor suites, domain differences, and the absence of camera input. Fig. 3 presents qualitative results from this experiment. Despite these challenges, our proposed ProFusion3D model demonstrates promising results.

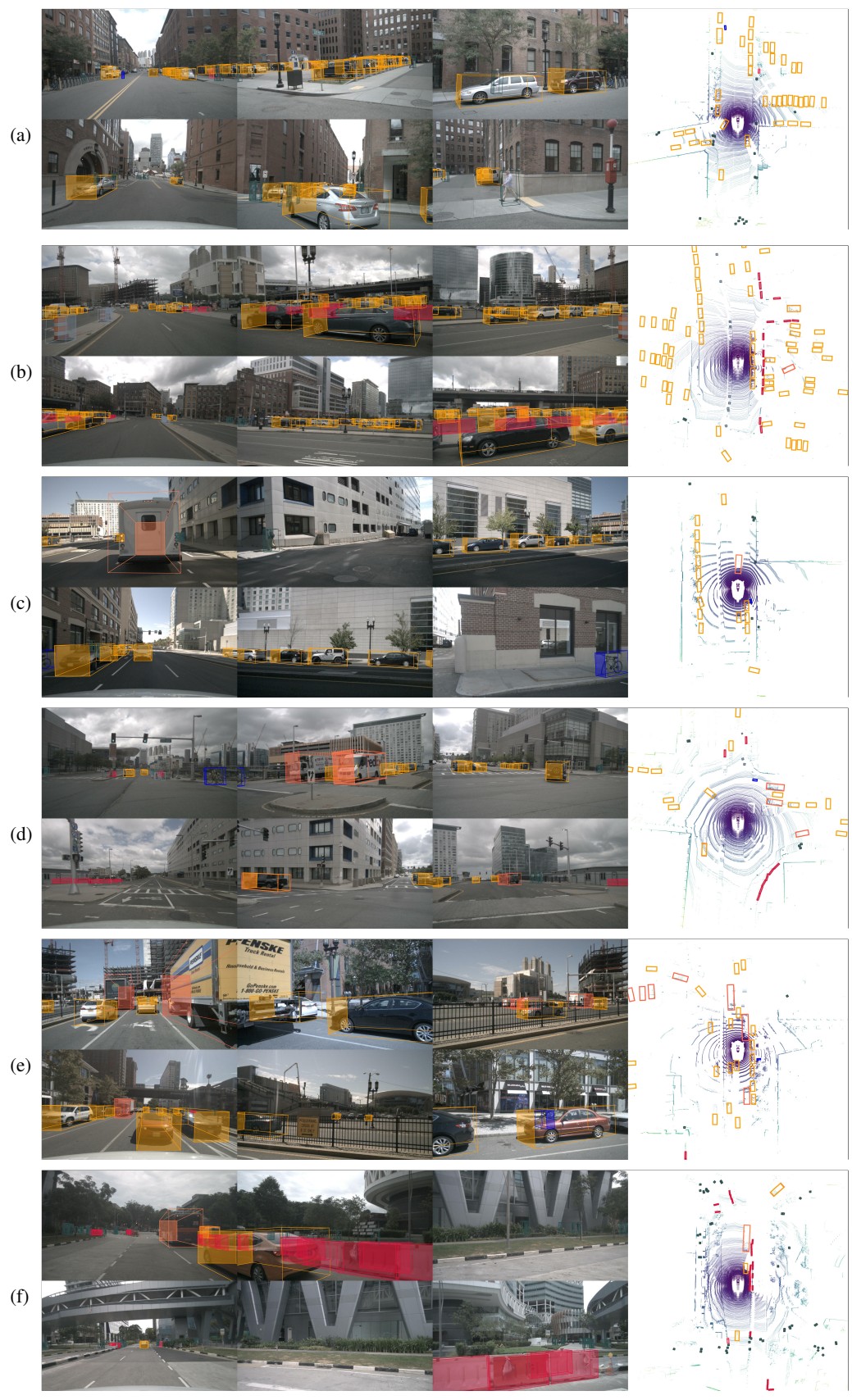

Figure 1: Visualization of 3D object detection prediction of our proposed ProFusion3D on the validation set of nuScenes. Classes are color-coded as follows: ■ car, ■ barrier, ■ truck, ■ cone, ■ bicycle, ■ person.

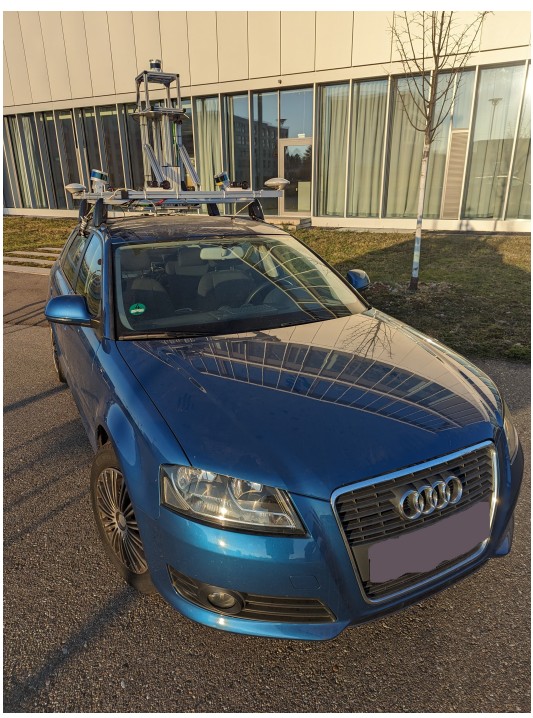

Figure 2: Illustration of our in-house autonomous driving vehicle used to demonstrate the robustness of the ProFusion3D architecture on real-world scenes

# References

[1] Y. Liu, T. Wang, X. Zhang, and J. Sun. Petr: Position embedding transformation for multi-view 3d object detection. In *European Conference on Computer Vision*, pages 531–548. Springer, 2022.

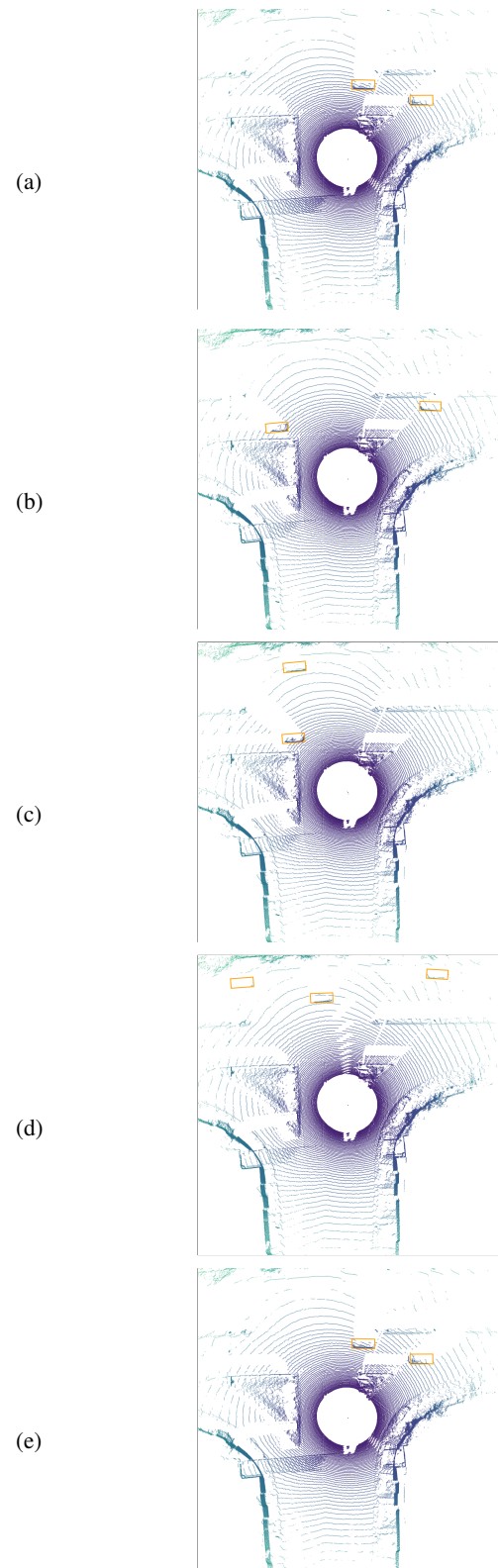

(a)

(b)

(c)

(d)

(e)

Figure 3: Visualization of 3D object detection predictions by our proposed ProFusion3D on real-world scenes, demonstrating its robustness under challenging conditions such as variations in sensor suites, domain differences, and the absence of camera images.

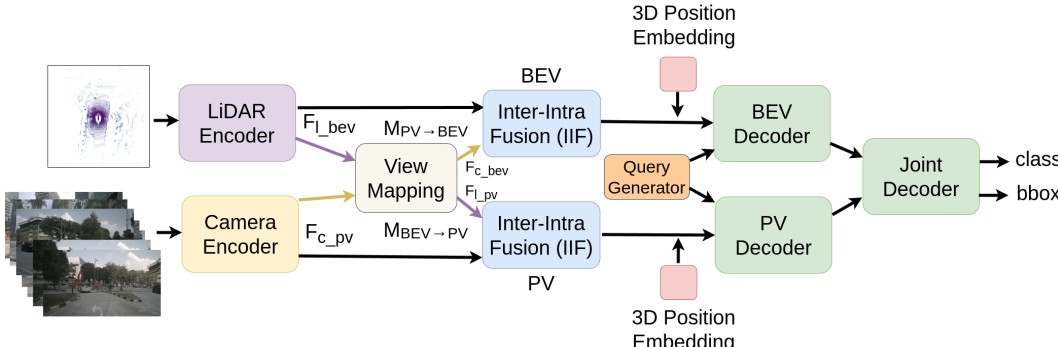

Figure 4: Illustration of our proposed ProFusion3D architecture that employs progressive fusion.

