# OpenReview forum: "Progressive Multi-Modal Fusion for Robust 3D Object Detection"
_robot-learning.org/CoRL/2024/Conference — CoRL 2024_

### Official Review · Reviewer_nzZg · 2024-07-06
**ProFusion3D:  Multi-View Sensor Fusion and 3D Positional Awareness for 3D Object Detection**

**Originality:** 4
**Technical Quality:** 4
**Clarity Of Presentation:** 3
**Potential Impact:** 4
**Recommendation:** 3
**Confidence:** 4

**Review:**

**Overview**

The architecture takes both LiDAR point clouds and multi-view camera images as input. It encodes these inputs using their corresponding encoders, resulting in BEV features for the LiDAR modality and PV features for the camera modality. These BEV features $F_{I_{bev}}$ for LiDAR and PV features $Fc_{pv}$ for the camera are termed intra-modality features in their respective spaces. The $F_{I_{bev}}$ features are projected to PV, resulting in inter-modality features $F_{I_{pv}}$ using a mapping function $\mathcal{M}_{BEV} \rightarrow PV$.

Similarly, the camera PV features $Fc_{pv}$ are mapped to BEV features $Fc_{bev}$ using the mapping function $\mathcal{M}_{PV} \rightarrow BEV$. This involves a 2D-to-3D view projection to construct camera BEV features from the camera PV features. This is the cross-view mapping stage.

For the fusion process, the inter-intra fusion block fuses intra-modality and inter-modality features in each feature space. This involves two units: the first focuses on capturing local context, while the second captures global spatial context. The fusion block applies a window attention mechanism, where queries $Q_{intra}$ are obtained from intra-modality features, and keys $K_{unified}$ and values $V_{unified}$ are derived from the concatenated intra and inter-modality features. By shifting the key and value features while keeping the query features fixed, the architecture creates cross-window connections, enhancing cross-modal attention.


**Strengths**:
-  The progressive fusion approach, involving both intermediate and object query levels across BEV and PV, allows effective leveraging of both local and global features.
- The use of joint self and cross window attention within the inter-intra fusion block is a novel method for combining multi-modal features.
- Overall it is a well-written paper and the results look good.

**Weaknesses**:
- The writing syle of the paper can be improved. The detailed description of the fusion mechanism, including cross-view mapping and hierarchical attention, described in section 3.1 ProFusion3D Architecture is complex and hard to parse. Simplifying or providing more intuitive explanations could help ease understanding.
- Figures 2(a) are helpful but incredibly small. I prefer to see a connection in notation in Section 3 to the diagram to better understand the components.
-  Potentially a missing baseline[1] ?



[1] DeepFusion: Lidar-Camera Deep Fusion for Multi-Modal 3D Object Detection -  https://openaccess.thecvf.com/content/CVPR2022/papers/Li_DeepFusion_Lidar-Camera_Deep_Fusion_for_Multi-Modal_3D_Object_Detection_CVPR_2022_paper.pdf

**Quality Of The Limitations Section:**

3

**Questions For Rebuttal:**

- Can you elaborate on the process of generating 3D positional embeddings from the camera and BEV meshgrids?

- How does the Hungarian algorithm contribute to the matching process between predicted and ground truth objects during training?

- Could you highlight the difference between ProFusion3D and DeepFusion referenced above? Since its also fusing LiDAR and Camera features.

**Robotics Focus:**

3

**Summary Of Paper:**

The paper introduces ProFusion3D, a framework that performs progressive LiDAR-camera sensor fusion. The model fuses features at two stages: the intermediate feature level and the object query level. Fusion occurs across both Bird’s Eye View (BEV) and Perspective View (PV), enabling the model to leverage both local and global features effectively.

**Summary Of Recommendation:**

This is a good paper, with decent contributions and the paper is well-written. Because I'm not super up to date with SOT in the field, I am recommending weak accept, but happy to revise based on other reviewers recommendations.

---

### Official Review · Reviewer_anjQ · 2024-07-18
**Innovative Fusion Approach for 3D Object Detection in Autonomous Driving**

**Originality:** 3
**Technical Quality:** 3
**Clarity Of Presentation:** 3
**Potential Impact:** 3
**Recommendation:** 3
**Confidence:** 3

**Review:**

Strengths:

1. Novel progressive fusion approach addressing limitations of single-view or single-stage fusion methods.
2. Innovative inter-intra fusion module capturing both local and global spatial context.
3. Promising self-supervised pre-training strategy improving data efficiency.
4. Comprehensive ablation studies providing valuable insights.
5. State-of-the-art results on nuScenes and Argoverse2 datasets.
6. Improved robustness in scenarios with sensor failures.

Weaknesses:

1. Limited discussion on real-time performance and computational requirements.
2. Absence of real-world testing on actual autonomous vehicles.
3. Insufficient exploration of temporal aspects in autonomous driving scenarios.
4. Limited discussion on safety implications of improved detection capabilities.
5. Insufficient analysis of generalization to diverse driving environments.
6. While consistent, performance improvements are rather small.

The paper presents a significant contribution to 3D object detection for autonomous driving. The ProFusion3D architecture effectively leverages complementary strengths of LiDAR and camera data, crucial for robust and accurate perception in autonomous driving.
The self-supervised pre-training strategy addresses the challenge of data efficiency, particularly relevant given the cost and time involved in collecting and annotating large-scale autonomous driving datasets.
The robustness analysis for sensor failure scenarios is commendable, addressing a critical challenge in deploying learning-based systems for safety-critical applications.

However, the paper has notable limitations:
* Lack of real-time performance analysis: The paper should discuss computational requirements and inference time, comparing speed-accuracy trade-offs with existing methods.
* Absence of real-world testing: Including results from actual autonomous vehicle tests would provide stronger evidence of practical applicability.
* Limited temporal considerations: Exploring how ProFusion3D could leverage temporal information would enhance its relevance to real-world scenarios.
* Insufficient safety discussion: The paper should elaborate on how improved detection translates to safer decision-making in autonomous driving.
* Limited generalization analysis: A discussion on performance in driving environments not represented in the datasets would be valuable.
* Minor performance improvements might not warrant the added complexity.

Addressing these limitations would significantly strengthen the paper's impact and practical relevance to autonomous driving applications.

**Quality Of The Limitations Section:**

2

**Questions For Rebuttal:**

1. What is ProFusion3D's computational complexity and real-time performance compared to existing methods?
2. Have you conducted tests on actual autonomous vehicles? If so, how do the results compare to dataset evaluations?
3. How might ProFusion3D be extended to leverage temporal information?
4. How does the improved detection performance translate to safer decision-making?
5. How well does ProFusion3D generalize to driving conditions not represented in the datasets?

**Robotics Focus:**

3

**Summary Of Paper:**

This paper introduces ProFusion3D, a novel approach for 3D object detection using progressive fusion of LiDAR and camera data for autonomous driving. The method fuses features in both Bird's Eye View and Perspective View spaces at multiple levels, introduces a new inter-intra fusion module, and proposes a self-supervised pre-training strategy. The approach achieves state-of-the-art results on nuScenes and Argoverse2 datasets.

**Summary Of Recommendation:**

ProFusion3D presents innovative ideas for multi-modal fusion in 3D object detection and demonstrates impressive results on automotive datasets. The progressive fusion approach, inter-intra fusion module, and self-supervised pre-training strategy are valuable contributions. However, the paper would be strengthened by addressing real-time performance, including real-world testing results, and exploring temporal aspects. I recommend minor revisions to address these points, after which the paper would make a strong contribution to the field of perception in autonomous driving.

---

### Official Review · Reviewer_9BjW · 2024-07-18
**Progressive Multi-Modal Fusion for Robust 3D Object Detection**

**Originality:** 4
**Technical Quality:** 4
**Clarity Of Presentation:** 5
**Potential Impact:** 4
**Recommendation:** 3
**Confidence:** 5

**Review:**

Strengths
* This paper proposes to fuse features at both intermediate and object query levels in both Bird's Eye View (BEV) and Perspective View (PV), aiming to leverage the strengths of different fusion stages and viewpoints.
* It's novel to propose multi-modal masked modeling in pre-training stage as it can leverage self-surpervised learning and information from the other modality to further boost the performance. To the best of my knowledge, it's the first time multi-modal masked modeling is used in 3d object detection
* This paper is clearly written, presenting a well-structured approach to addressing the challenges of LiDAR-camera fusion for 3D object detection in autonomous driving scenarios. The authors articulate their methodology logically, progressing from the problem statement through their proposed solution and its implementation.
* The ablation study is very thorough, covering all the technical contributions studied in this paper. It systematically examines each component of the ProFusion3D framework, including the progressive fusion approach, the inter-intra modality fusion module, and the self-supervised pre-training strategy. This comprehensive analysis helps to validate the effectiveness of each proposed element and provides insights into their individual and combined impacts on the overall performance of the 3D object detection system

Weaknesses
* The performance gaps between ProFusion3D and other SOTA methods are limited, especially comparing to UniTR.
* I am having trouble understand why the performance of ProFusion3d in Table 1a doesn't match that of Table 2a
* In the single sensor setting, ProFusion is worse than BEVFormer and TransFusion-L even with pre-training.

**Quality Of The Limitations Section:**

3

**Questions For Rebuttal:**

* For the data efficiency experiment in section 4.2, it's not very cleary that the model are 1. pre-trained on subset of training data and finetuned on subset  or 2. pre-trained on full set of training data and and finetuned on subset.
* It would be interesting to see how the performance change if the model is pretrained on single modelity

**Robotics Focus:**

3

**Summary Of Paper:**

The paper introduces ProFusion3D, a novel framework for LiDAR-camera fusion in 3D object detection for automated driving, which employs a progressive fusion approach across multiple stages and views.

**Summary Of Recommendation:**

Even though the performance increase on top of SOTA methods is not significant, the novelty of this paper, especially multi-modal mask modeling, lead to my recommendation of this paper

---

### Author Rebuttal · Authors · 2024-08-14

We sincerely appreciate the thoughtful feedback and valuable comments from all the reviewers. We have carefully revised the paper to address all the concerns and suggestions raised. We hope that the reviewers recognize the value in our work and consider raising their scores. Thank you for your invaluable time and consideration.

The additional rebuttal experiments are presented in the official comment titled **"Additional Experiments for Rebuttal"**. Below, we outline the changes made in the updated manuscript based on the reviewers' feedback. The revisions in the manuscript are marked in blue text, which is included in the rebuttal.zip.

1. [9BjW] In Section 4.2 Ablation Study, we have outlined the experimental settings for Tables 2a and 2b. (Page 7)

2. [9BjW] In the data efficiency experiment discussed in Section 4.2 of the Ablation Study, we clearly state that pre-training is done on the entire dataset. (Page 7)

3. [anjQ] We have added a brief discussion on the importance of 3D object detection for safer decision-making. (Page 1)

4. [nzZg] We have re-written the cross-view mapping and Inter-Intra Fusion segments of Section 3 to improve clarity. (Pages 3 and 4)

5. [nzZg] We have added Fl_bev, Fl_pnev, Fc_pv, and Fc_bev to Figure 2(a) to improve its connection with Section 3. (Page 3)

6. [nzZg] We have included DeepFusion in the related work section. (Page 2)

---

### Decision · Program_Chairs · 2024-09-04

**Decision:**

Accept

**Comment:**

Strengths:
- a novel idea;
- detailed ablations;
- clear presentation.

Weaknesses:
- limited discussion on real-world applications;
- marginal performance gain;
- some missing technical details.

Post-rebuttal: The authors did an impressive job during the rebuttal, with a significant number of additional experimental results.  All three reviewers recommended acceptance, and the AC agreed.  The authors are encouraged to include the content from the rebuttal in the camera-ready version.